

# How intermittency affects the rate at which rainfall extremes respond to changes in temperature

Schleiss Marc[1]

[1]Department of Geoscience & Remote Sensing, Delft University of Technology

**Correspondence:** Marc Schleiss (m.a.schleiss@tudelft.nl)

**Abstract.** A detailed analysis of how intermittency modulates the rate at which sub-daily rainfall extremes depend on temperature is presented. Results show that hourly extremes tend to be predominantly controlled by peak intensity, increasing at a rate of approximately 7% per degree in agreement with the Clausius-Clapeyron equation. However, rapid increase of rainfall intermittency upward of 20-25°C is shown to produce local deviations from this theoretical scaling, resulting in lower correlations between rainfall and temperature. On the other hand, rapidly decreasing intermittency with temperature between 10-20 degrees can result in higher net scaling rates than expected, potentially exceeding Clausius-Clapeyron. In general, the importance of intermittency in controlling the scaling rates of precipitation totals with temperature grows as we progress from hourly to daily aggregation time scales and beyond. Thermodynamic effects still play an important role in controlling the maximum water holding capacity of the atmosphere and therefore peak rainfall intensity. But the observational evidence shows that beyond a few hours, storm totals become increasingly dominated by dynamical factors such as event duration and internal intermittency. The conclusion is that Clausius-Clapeyron scaling alone can not be used to reliably predict the net effective changes in rainfall extremes with temperature beyond a few hours. A more general scaling model that takes into account simultaneous changes in intermittency and peak intensity with temperature is proposed to help better disentangle these two phenomena (e.g., peak intensity and intermittency). The model is applied to a large number of high-resolution rain gauge time series in the United States. Results show that the new model with intermittency greatly improves the representation of rainfall extremes with temperature, producing a much more consistent and reliable picture of extremes across scales than what can be achieved using Clausius-Clapeyron only.

## 1 Introduction

Recently, there has been an increased interest in understanding and predicting changes in precipitation extremes due to global warming (e.g., Trenberth et al., 2003; Frei et al., 2006; Allan and Soden, 2008; Trenberth, 2011; Muschinski and Katz, 2013; Westra et al., 2014; Ban et al., 2015; Groisman et al., 2015; Donat et al., 2016; Scherrer et al., 2016). Most studies on this topic agree that heavy rainfall is likely to increase in the future. However, a clear framework for predicting changes in rainfall



intensities across scales is still lacking. The general consensus seems to be that in places with sufficient moisture availability, rainfall extremes will increase at the same rate as the moisture holding capacity of the atmosphere, that is, at a rate of about 7% per degree of warming in accordance to the Clausius-Clapeyron relationship. The observational evidence, however, points to a more complicated picture (Shaw et al., 2011). It shows that rainfall extremes and temperatures are linked in more complicated

ways, depending on the local climatology, moisture availability, large-scale forcing, orography and scale of analysis (e.g., Panthou et al., 2014; Donat et al., 2016; Drobinski et al., 2016; Ivancic and Shaw, 2016; Barbero et al., 2017). Some hourly rainfall extremes, for example, have been shown to increase at rates of up to 14% per degree Celsius – about twice as fast as expected from Clausius-Clapeyron (Lenderink and van Meijgaard, 2008; Lenderink et al., 2011). Haerter and Berg (2009) and Berg et al. (2013) argue this is due to fundamental differences in scaling between large-scale stratiform extremes, which are

expected to increase with temperature at the Clausius-Clapeyron rate, and small-scale convective extremes which may exceed Clausius-Clapeyron due to dynamical processes like strong local updrafts and downdrafts. Lepore et al. (2015) argue along the same line, highlighting the importance of separating the effects of temperature on rainfall extremes via increased atmospheric water content (described by Clausius-Clapeyron) and via enhanced atmospheric convection and moisture convergence.

One important yet poorly discussed issue in all these studies concerns the role played by intermittency (i.e., the alternation

of dry and rainy periods) in controlling the response of rainfall extremes to changing temperatures. Indeed, beyond a few hours of aggregation time scale, total rainfall amounts often turn out to be more correlated to storm duration and intermittency rather than peak rainfall intensity (e.g., Azad and Sorteberg, 2017; Lamjiri et al., 2017). Because intermittency itself might be sensitive to changes in temperature, Clausius-Clapeyron scaling alone might not be sufficient to fully understand the relation between rainfall extremes and temperature (Haerter et al., 2010; Panthou et al., 2014; Wasko et al., 2015).

A recent study by Wasko et al. (2016) found that although peak rainfall intensity usually scales positively with temperature, event duration and spatial extent tend to decrease with temperature. Global climate model simulations by Dwyer and O'Gorman (2017) partly confirm this trend, projecting a global 1% decrease in extreme precipitation duration per degree of warming. However the magnitude of this trend remains highly uncertain due to strong limitations in the ability of global climate models to simulate realistic intermittency patterns. A recently proposed idealized stochastic model by Neelin et al. (2017) gives additional

insight into the main mechanisms responsible for controlling the complicated interaction between event duration, peak intensity and total rainfall amount. It shows that mathematically, the problem can be represented as a competition (over time) between moisture convergence and water losses by precipitation. While too idealized, the model still highlights some important aspects like the fact that unless there is rapid replenishment of moisture from surrounding regions (e.g., through advection), big storms with heavy precipitation at their core will tend to run out of moisture more quickly, forcing precipitation to cease earlier

(Trenberth et al., 2003). The consequence of this is a more complicated scaling pattern in which rainfall extremes do not necessarily increase at the same rate for all temperatures and across all scales of aggregation. Temperature still plays a crucial role by limiting the maximum water holding capacity of the air and controlling evaporation rates in surrounding regions. However, dynamical factors like intermittency also need to be considered.

The present study aims at shedding new light on this important issue by presenting a detailed statistical analysis of how

intermittency modulates the rate at which precipitation extremes (in current climate) depend on temperature. Results show





that at time scales of 1 h or less, rainfall extremes tend to be predominantly controlled by changes in temperature. However, rapid local increases or decreases in intermittency with temperature can significantly lower or amplify the net scaling rates. In extreme cases, this may lead to (locally) negative scaling rates or, conversely, super-Clausius Clapeyron scaling. As we move towards rainfall extremes at daily scales and beyond, intermittency rapidly gains in importance, masking most of the

thermodynamic effects. To disentangle the two, a more general scaling model that takes into account simultaneous changes in intermittency and maximum intensity with temperature is proposed. Results show the new model greatly improves predictions of rainfall extremes with temperature, producing a more consistent and reliable depiction of observed responses across a wide range of temporal scales.

    The rest of this paper is structured as follows: Section 2 introduces the data used for the analysis. Section 3 describes the

methods and models used to detect and analyze rainfall extremes. The main results are provided in Section 4. The first part focuses on how intermittency affects the scaling of rainfall extremes across temperatures and time scales. The second part analyzes the goodness of fit of the newly proposed model and the third and last part investigates the sensitivity of derived scaling rates with respect to the chosen quantile $q$. The conclusions and some additional ideas for future research are given in Section 5.

## 15   2   Data

The data used in the study were taken from the sub-hourly U.S. Climate Reference Network (Diamond et al., 2013). Two main quantities were considered: total precipitation depth $R$ (expressed in $mm$) and average air temperature $T$ (in Celsius). Initial analyses also included dew point temperature $T_d$ (in Celsius), estimated by combining relative humidity and air temperature using the Magnus formula (e.g., Alduchov and Eskridge, 1996). However, dew point time series were too short to derive

reliable results and it was decided to focus only on air temperature.

    The raw rainfall and temperature time series had constant 5-min temporal resolution and spanned 232 locations in the United States, Canada and Siberia from 2006 to August 2017. However, only the stations with sufficiently long data records were kept (see below). Note that the 5-min temperature values were derived by averaging independent 10 second measurements from multiple co-located sensors. Similarly, precipitation totals (both solid and liquid) were estimated using a weighing bucket

gauge with three independent load cell sensors. All 5-min measurements that failed automatic quality control tests were flagged and removed prior to analysis.

    Since the goal of this paper is to analyze the properties of rainfall extremes across different scales, all time series were aggregated from their original resolution of 5 min to larger time scales of 1 h, 2 h, etc... up to 24 h in regular steps of 1 h. Aggregation was performed over overlapping time windows (shifted by 5 min), taking the sums of all 5-min rainfall amounts

in each time interval. Air temperature was aggregated using the arithmetic mean and values were binned into regular classes of 1 degree Celsius. All aggregation time windows containing one or several missing values were discarded prior to analysis. Only the stations with at least 20 valid positive rainfall values in at least 20 different temperature classes between 5 and 30 degrees Celsius at the 24 h aggregation time scale were kept for the analysis. This drastically reduced the number of stations,





from 232 to 99. A map with the 99 stations satisfying all these criteria is shown in Figure 1. For illustration purposes, one randomly selected station (i.e., AL-Fairhope-3-NE) in the southern part of the country was selected (see red cross in Figure 1). The station is representative of a humid subtropical climate with plenty of moisture availability and a mixture of both small and large-scale rainfall extremes.

## 3 Methods

### 3.1 Definition of Rainfall Extremes

Consider a time series of strictly positive rainfall amounts $R_1(\Delta t), \ldots, R_N(\Delta t)$ at temperatures $T_1(\Delta t), \ldots, T_N(\Delta t)$. Each $R_i(\Delta t)$ represents the total accumulated rainfall amount (in mm) over a time period of length $\Delta t > 0$. The couples $(R_i, T_i)$ can be seen as realizations of a bivariate random variable $(R_{\Delta t}, T_{\Delta t})$ with joint probability distribution function $F_{\Delta t}$. A rainfall amount $R_i(\Delta t) > 0$ at time scale $\Delta t$ and temperature $T_i(\Delta t) = T$ is said to be "extreme" if it exceeds the $q^{\text{th}}$ quantile $R_q(\Delta t, T)$ of all strictly positive rainfall amounts at this temperature and aggregation time scale:

$$\mathbb{P}[R_{\Delta t} \leq R_q(\Delta t, T) \mid T_{\Delta t} = T) = q \tag{1}$$

where $\mathbb{P}$ denotes the probability and $0 < q < 1$ is the quantile of interest. To have enough observations, temperature measurements are binned in 20 different classes between 5 and 30 degrees Celsius. Given the relatively small sample sizes (for each temperature class), the default quantile used in this paper is $q = 0.95$. However, 50 other values of $q$ between 0.95 and 0.999 (in regular steps of 0.001) are also considered for the sensitivity analysis in the last part of the paper. To avoid large estimation errors, quantile $q > 0.95$ was computed only if there were at least $\lceil \frac{1}{1-q} \rceil$ strictly positive rainfall observations in a given temperature class, with $\lceil x \rceil$ denoting the upper integer part of $x$. Consequently, not all 50 different quantiles are available for each of the 99 selected stations.

### 3.2 Internal Intermittency

Consider an aggregated rainfall amount $R_i(\Delta t)$ at scale $\Delta t > 0$ with $\Delta t = n \cdot \Delta t_0$. The variable $n \in \mathbb{N}$ represents the aggregation ratio with respect to a smaller observation scale $\Delta t_0$. For example, $\Delta t_0 = 5$ min and $n = 12$ for hourly aggregated 5-min rainfall amounts and $n = 288$ for daily aggregated 5-min amounts. By definition, each $R_i(\Delta t)$ can be expressed as the sum of $n$ individual observations at smaller scale $\Delta t_0$:

$$R_i(n\Delta t_0) = \sum_{j=ni-n+1}^{ni} R_j(\Delta t_0) \tag{2}$$

The fraction of dry periods at reference scale $\Delta t_0$ contained within $\Delta t$ is called the internal intermittency of $R_i(\Delta t)$ and is denoted by:

$$I_i(\Delta t) = \frac{1}{n} \sum_{j=ni-n+1}^{ni} \mathbb{1}_{\{R_j(\Delta t_0)=0\}} \tag{3}$$





where $\mathbb{1}_{\{x\}}$ is a function that equals 1 if $x$ is true and 0 otherwise. Note that for more conciseness, the reference scale $\Delta t_0$ has been omitted in the notation for $I_i(\Delta t)$. However it should be clear from the definition that $I_i(\Delta t)$ is a *relative* measure of lacunarity with respect to a fixed reference scale $\Delta t_0$. The smaller this reference scale, the larger the internal intermittency. Fortunately, this scale-dependence is not a major problem here as we are mostly interested in understanding relative changes

in intermittency from one temperature to another.

Also, note that because of the original sampling resolution of 5 min in the USCRN data, the internal intermittency of a rainfall amount at scale $n \cdot \Delta t_0$ can only be estimated with an accuracy of at best $1/n$. Small-scale intermittency estimates are therefore affected by relatively strong discretization effects that could potentially mask the changes from one temperature to another. To mitigate this effect, the smallest aggregation time scale considered in this paper will be 60 min ($n = 12$) which means that the

maximum uncertainty affecting intermittency estimates is 8.3%. However, uncertainty decreases with aggregation time scale, down to 0.347% at the daily time scale.

### 3.3 Intermittency of Rainfall Extremes

The way rainfall intermittency varies with spatial and temporal aggregation scale has already been been studied quite extensively (e.g., Kumar and Foufoula-Georgiou, 1994; Jeannin et al., 2008; Kundu and Siddani, 2011; Schleiss et al., 2011; Mascaro

et al., 2013; Dunkerley, 2015). So far, however, very few studies have tried to characterize the conditional expectation of intermittency at a given rainfall intensity, temperature and scale. The latter plays a crucial role in many applications, including flood forecasting, radar remote sensing and the design of stochastic rainfall simulators capable of preserving the structure and dynamics of rainfall across scales.

Figure 2 shows the 4 largest observed rainfall accumulations at the 1 h, 2 h, 6 h and 24 h aggregation time scales for the

station in Fairhope, Alabama. As can be expected, rainfall amounts steadily increase with aggregation scale. Average rainfall intensities on the other hand decrease from 95 mm h$^{-1}$ at the hourly time scale to less than 13.6 mm h$^{-1}$ at the daily scale. A large part of this seven-fold decrease can be attributed to the strong increase in internal intermittency, from roughly 0% at hourly time scale to more than 64% at the daily time scale, highlighting the fundamentally different physical processes through which extreme rainfall accumulations at small and large time scales are produced. In the following, we explain the methodology used

in this paper to generalize this type of analysis to other rainfall quantiles and study variations in intermittency across scales and temperatures.

Similarly to rainfall and temperature, it is possible to represent the internal intermittency $I_i(\Delta t)$ of rainfall accumulations at a given time scale $\Delta t$ as realizations of a random variable $I_{\Delta t}$. Combining all the variables together produces triplets $(R_{\Delta t}, T_{\Delta t}, I_{\Delta t})$ of simultaneous rainfall accumulations, temperatures and internal intermittency values. Detailed analysis of the

joint and marginal distributions of this trivariate random function is necessary to fully understand the link between rainfall extremes and temperature across scales. Unfortunately, due to the short data record, this proves very challenging (especially for extremes). A simpler approach is to focus on the expected intermittency $I_q(\Delta t, T)$ of the q$^{\text{th}}$ rainfall quantile $R_q(\Delta t, T)$



conditionally on temperature $T$:

$$I_q(\Delta t, T) = \mathbb{E}[I_{\Delta t} \mid R_{\Delta t} = R_q(\Delta t, T) \text{ and } T_{\Delta t} = T] \tag{4}$$

In this study, expected intermittency values (conditional on temperature) were estimated by fitting a logistic regression between the internal intermittency and the logit of $q$, as shown in Figure 3:

$$g(I_q(\Delta t, T)) = c_0(\Delta t, T) + c_1(\Delta t, T)g(q) \tag{5}$$

with the logit function $g(x)$ given by:

$$g(x) = \ln\left(\frac{x}{1-x}\right) \tag{6}$$

The two model parameters $c_0(\Delta t, T)$ and $c_1(\Delta t, T)$ are fitted numerically for each of the selected stations, temperature $T$ (between 5 and 30°C) and aggregation time scale $\Delta t$ between 1 h and 24 h. The fit is performed using the glm() function in

the statistical programming language R (R Core Team, 2017). The fitted model parameters $\hat{c}_0(\Delta t, T)$ and $\hat{c}_1(\Delta t, T)$ at each temperature $T$ can then be used to estimate the expected internal intermittency of extremes for any given rainfall quantile $q$:

$$\hat{I}_q(\Delta t, T) = g^{-1}\left(\hat{c}_0(\Delta t, T) + \hat{c}_1(\Delta t, T)g(q)\right) \tag{7}$$

with

$$g^{-1}(x) = \frac{e^x}{e^x + 1} \tag{8}$$

## 3.4  Scaling of Extremes

Scaling analyses in this paper are performed by considering the mean air temperature $T_i(\Delta t)$ over the same aggregation time period than the rainfall amounts $R_i(\Delta t)$. Some previous studies suggested to use temperatures recorded immediately before/after the rain event to avoid potential "contamination" by the rain itself. However, averaging temperatures before/after an event is questionable for at least two reasons. Firstly, the start/end of an event are often very subjective and highly dependent on

the scale of analysis. And secondly, prior/posterior temperatures may not necessarily have the same representativity depending on when rain occurred and how long it lasted. For these reasons, only simultaneous temperature and rainfall measurements will be used.

Previous studies have shown that rainfall extremes $R_q(\Delta t, T)$ for large $q$ increase approximately exponentially with temperature $T$:

$$R_q(\Delta t, T) \approx e^{\alpha_q(\Delta t) + \beta_q(\Delta t)T} \tag{9}$$

where $\alpha_q(\Delta t) \in \mathbb{R}$ and $\beta_q(\Delta t) \in \mathbb{R}$ are two parameters depending on the quantile $q$ and aggregation scale $\Delta t$. The exponential relationship is (partly) justified by the fact that saturation water vapor pressure exponentially grows with temperature, meaning





that under conditions of constant relative humidity, precipitable water should scale approximately similarly (Trenberth et al., 2003). Assuming model (9) holds, the scaling rate $\lambda_q(\Delta t)$ per unit increase in temperature $T$ is given by:

$$\lambda_q(\Delta t, T) = \frac{R_q(\Delta t, T+1)}{R_q(\Delta t, T)} - 1 = e^{\beta_q(\Delta t)} - 1 \tag{10}$$

which is a constant and does not depend on $T$. One of the main problems with the constant scaling assumption above is that

it only seems to hold in approximation and over a limited range of temperatures. In particular, departures from log-linearity have been reported at temperatures below 5-10°C and above 20-25°C (e.g., Lenderink and van Meijgaard, 2008; Haerter et al., 2010). One way to account for these deviations and increase robustness is to take the log transform of rainfall quantiles and derive the slope parameter $\beta_q(\Delta t)$ in Equation (9) using the non-parametric Theil-Sen estimator (Sen, 1968) instead of traditional least squares. Even so, average scaling rates derived using this technique may not be very representative of the actual

changes in extreme rainfall amounts across the whole temperature range.

To address this limitation, another slightly more general scaling model is proposed in which a multiplicative correction term is added in Equation (9) to account for possible changes in intermittency with scale and temperature:

$$R_q(\Delta t, T) \approx [1 - I_q(\Delta t, T)] \cdot e^{\alpha_q^{\text{new}}(\Delta t) + \beta_q^{\text{new}}(\Delta t)T} \tag{11}$$

where $I_q(\Delta t, T)$ is the expected internal intermittency of rainfall extremes exceeding the $q^{\text{th}}$ quantile (conditionally on tem-

perature $T$) and $\alpha_q^{\text{new}}(\Delta t)$ and $\beta_q^{\text{new}}(\Delta t)$ are two new model parameters. The idea is similar to that proposed by Wasko et al. (2015) in which the authors related peak rainfall intensity to temperature with a correction for storm duration. However, the parametric form of their model was slightly different from the one proposed here. Also, the main covariate modulating the rainfall amounts in (11) is internal intermittency and not rainfall duration.

Since the internal intermittency $I_q(\Delta t, T)$ in Equation (11) changes with temperature, relative rates of increase/decrease in

rainfall amounts per unit change in temperature are not independent of $T$ anymore like in Equation (10) but modulated by local changes in intermittency:

$$\lambda_q^{\text{new}}(\Delta t, T) = \frac{1 - I_q(\Delta t, T+1)}{1 - I_q(\Delta t, T)} e^{\beta_q^{\text{new}}(\Delta t)} - 1 \tag{12}$$

As a result, rainfall amounts can either increase or decrease with temperature, leading to a better differentiation between the thermodynamic effects (i.e., increase in moisture holding capacity with temperature) and dynamic effects caused by changes

in intermittency with temperature. In fact, since Equation (11) is equivalent to a renormalization of the rainfall amounts by a factor $1 - I_q(\Delta t, T)$, we can retrieve the "true" underlying scaling rate with temperature after correction for intermittency:

$$\frac{R_q(\Delta t, T)}{1 - I_q(\Delta t, T)} \approx e^{\alpha_q^{\text{new}}(\Delta t) + \beta_q^{\text{new}}(\Delta t)T} \tag{13}$$

$$\lambda_q^{\text{true}}(\Delta t) = e^{\beta_q^{\text{new}}(\Delta t)} - 1 \tag{14}$$

where $\lambda_q^{\text{true}}(\Delta t)$ represents the scaling rate with temperature for the intermittency corrected rainfall amounts. The goal of this paper is to study these scaling rates with temperature and intermittency across different regions and quantify their relative importance for a wide range of temporal aggregations scales.





## 4   Results

### 4.1   The Effect of Intermittency on the Scaling of Rainfall Amounts

Figure 4 shows the 95th quantile of hourly rainfall accumulations at Fairhope as a function of air temperature, together with the corresponding intermittency estimates from the fitted generalized linear model in Equation 5. The average rate of increase with
temperature, as indicated by the black dotted line, is 3.2% per degree Celsius. However, there appears to be two distinct scaling patterns with temperature. The first, between 5° and 20°C is characterized by a steady increase in rainfall amounts of 6.5% per degree Celsius, in good agreement with the Clausius-Clapeyron relationship. The second part, from 20° to 30°C, exhibits a negative trend in rainfall amounts of -4.5% per degree Celsius associated to a sudden and rapid increase in intermittency. This is consistent with previous evidence presented by Lenderink et al. (2011) and Berg et al. (2013) who pointed out similar
changes in scaling above 22-23°C. Lenderink et al. (2011) could not fully explain the reasons behind this but suggested it could be due to micro-physical processes occurring in convective clouds. Berg et al. (2013) argue along a slightly different line. Their working hypothesis is that convective precipitation extremes scale much faster with temperature than stratiform extremes. The change in scaling rate at higher temperatures could therefore be explained by relative changes in the frequencies of stratiform over convective precipitation extremes beyond 22-23°C.

The explanation proposed here is much simpler: rainfall extremes at higher temperatures tend to be more intermittent than at lower temperatures, modifying the local rate at which the rainfall quantiles vary with temperature. A simple model with fixed scaling rate as a function of temperature is incapable of reproducing such variations. The modified scaling model in Equation (11) shown in the top panel of Figure 4 on the other hand, performs much better. It correctly reproduces the observed decrease in rainfall quantiles at higher temperatures without the need to separate stratiform from convective events. Most
importantly, it shows that once intermittency has been accounted for, as detailed in Equation (14), a clear and consistent positive trend of rainfall intensity of approximately 6.6% per degree Celsius over the entire range of temperatures emerges. This 6.6% increase represents the "true" underlying trend with temperature on top of which an additional (non-linear) component due to intermittency can be added.

Figure 5 shows the same type of analysis for the station in Fairhope but this time at the daily aggregation scale. The top
panel shows the effective scaling rate with temperature is close to zero, meaning that temperature alone is not a good predictor of rainfall accumulations at larger time scales. Intermittency on the other hand, appears to exert a much stronger control over the total accumulation, as indicated by the strong rank correlation coefficient of -0.64. Similarly to the hourly scale, there appears to be a sudden and rapid increase in internal intermittency at temperatures above 20°C. The modified scaling model accounting for intermittency performs much better, predicting an increase in rainfall amounts of 5.6% per degree Celsius. This
is slightly smaller than Clausius-Clapeyron scaling but still reasonable given that we only considered the 95th quantile of all rainfall amounts.

Similar analyses of the 95th rainfall quantiles and intermittency for all 99 stations in the dataset in Figures 6 and 7 confirm this general scaling pattern. At the hourly scale and temperatures below 20°C, the median intermittency of extremes tends to be very low. Temperature therefore naturally tends to play a much more important role in influencing rainfall amounts. However,



as we move toward larger scales and higher temperatures, intermittency progressively gains in importance. The exact scale at which intermittency starts to exert more control over total amounts than temperature depends on the considered station. But overall, the transition usually occurs at temporal aggregation scales of 3 h to 6 h.

Figure 8 provides more insight into how intermittency affects the scaling of rainfall extremes with temperature and aggregation time scale. In the model without intermittency, scaling rates rapidly decrease with $\Delta t$ from approximately 4.37% at the hourly time scale to -0.45% at the 24 h scale. The rapid decrease in scaling rate conveys the wrong idea that extremes at larger scales do not depend on temperature. However, this is an artifact caused by rapidly increasing intermittency at higher temperatures and aggregation time scales. In other words, intermittency is a confounding factor affecting the scaling of rainfall totals with temperature. After correcting for it, the effect of temperature becomes visible again. Still, there appears to be a small decrease of the scaling rate with $\Delta t$ from 8.0% at the hourly scale to 5.70% at the 24 h scale which might be due to the relatively small sample sizes. In general, however, the intermittency corrected scaling rates are much closer to what can be expected from the Clausius-Clapeyron relationship.

The stations with the strongest scaling rates overall (both at the hourly and daily time scales) were FL-Sebring-23-SSE (12.96% without intermittency and 14.70% with intermittency) and FL-Everglades-City-5-NE (12.42% respectively 13.04%), both situated in a humid tropical climate famous for large and intense warm season thunderstorms. Apart from these two, no other station exhibited scaling rates in excess of 12% per degree Celsius. In general, we observe that the lowest scaling rates with temperature (both corrected and uncorrected for intermittency) tend to be associated with moisture limited places (e.g., CA-Stovepipe-Wells-1-SW, CA-Fallbrook-5-NE, UT-Brigham-City-28-WNW and NM-Clayton-3-ENE). The state of California is a particularly interesting case. Uncorrected scaling rates at CA-Fallbrook-5-NE for example, were 0.07% at the hourly time scale and -10.42% at the daily time scale. The strong negative scaling rate at the daily time scale can be explained by the fact that, unlike the southern and central parts of the United States, large-scale precipitation extremes along the West Coast usually occur during the cold season. They are associated with rapid transport of moisture from the Pacific ocean towards the mainland along atmospheric rivers which results in a very steady and persistent rain over time (Berg et al., 2002; Bracken et al., 2015; Lamjiri et al., 2017). The scaling model that corrects for changes in intermittency removes the negative trend with temperature. But even the corrected scaling rates remain relatively low at 1.67% for the hourly scale and 2.24% at the daily scale, confirming that large scale moisture transport and storm dynamics play a much more important role than temperature in determining rainfall totals over this region of the globe.

Overall, the results confirm that air temperature alone is not systematically a good indicator for understanding extreme rainfall accumulations, and conditions in surrounding regions must be taken into account as well. The correction for intermittency makes it easier to understand and characterize the "true" sensitivity of heavy rainfall to changes in air temperatures across scales and geographical regions. But significant uncertainty remains and the corrected model does not tell the full story either. However, it offers new insight into the nature of rainfall extremes which are helpful in explaining some of the abnormally low/high scaling rates that we see in the observational record.



## 4.2 Goodness of fit

Repeating the same type of analysis as above, we computed the root mean square error (RMSE) and coefficient of determination ($R^2$) of the two different scaling models (i.e., with/without intermittency) for all 99 stations in the dataset and across all scales of aggregation (see Figure 9). On average, the model that corrects for intermittency (on the right) reduced RMSE values

by a factor 1.6 while increasing the coefficient of determination by 0.6. It systematically outperformed the simpler model based on temperature alone and kept a reasonable goodness of fit across all aggregation time scales, especially at the larger ones where temperature alone performed poorly. The 5 biggest improvements in model performance at the hourly time scales were observed in the southern states at MS-Holly-Springs-4-N, AL-Russellville-4-SSE, LA-Lafayette-13-SE, LA-Monroe-26-N and SC-McClellanville-7-NE, all characterized by large moisture availability and rapid increases in intermittency at

higher temperatures. The 5 largest improvements at the 24 h time scale on the other hand were located in more moisture limited places such as NM-Artesia-2-WNW, NM-Vaughn-36-SSE, TX-Port-Aransas-32-NNE, TX-Austin-33-NW and UT-Blanding-26-SSW. In these regions, large storm totals strongly depend on intermittency and steady transport of moisture from surrounding regions.

The comparisons above show that while temperature plays an important role in shaping rainfall extremes at smaller scales,

its effects at larger scales are likely to be masked by changes in storm dynamics, such as increased intermittency. Additional correlation analyses between the 95th rainfall quantile and internal intermittency with temperature presented in Figure 10 provide more insight into this phenomena. They show that the median rank correlation between amounts and intermittency decreases from 0.12 at the hourly scale to -0.48 at the daily scale. The fact that extremes at small aggregation scales below 3 h tend to be slightly positively correlated with intermittency is in agreement with the findings of Wasko et al. (2016). It

means that small-scale rainfall extremes at higher temperatures tend to be more concentrated in space and time while rainfall extremes at scales above a few hours tend to be associated with longer-lasting systems like the passage of a cold front or a system of thunderstorms in which a series of convective cells repeatedly moves over the same region. The effect of temperature on total rainfall amounts in this case becomes less clear, as large accumulations can occur both at low and high temperatures. This interplay between temperature, peak intensity, intermittency and storm totals outlines a more complicated picture than

traditionally depicted in Clausius-Clapeyron scaling analyses. It shows that rainfall extremes vary with temperature in ways that can not be fully explained by Clausius-Clapeyron but requires a more in-depth understanding of storm type, organization and dynamics. It also underlines why the ability to produce realistic storm dynamics and rainfall structures in global and regional climate models is so important for making credible projections about the future of rainfall extremes across scales.

## 4.3 Sensitivity to choice of quantile

So far, all results we have shown were for extremes exceeding the 95[th] quantile of rainfall accumulations. The goal of this last section is to quantify the sensitivity of the retrieved scaling rates with respect to the choice of the quantile $q$ used to identify the rainfall "extremes" in the first place. Previous studies have shown that relatively high quantiles are necessary in order to observe Clausius-Clapeyron scaling of rainfall extremes (Shaw et al., 2011). But a clear and detailed study of the influence of





$q$ on observed scaling rates is still missing. The reason the choice of quantile is important for scaling analyses is that higher rainfall amounts naturally tend to be associated with lower intermittency levels. This makes them more likely to scale with temperature. However, since intermittency might not change uniformly with $q$ and $T$, changes in scaling might not be obvious to anticipate.

Figure 11 shows box plots of the estimated scaling rates of rainfall amounts with temperature at the 1 h and 24 h aggregation time scales as a function of the considered quantile $q$. In the first model (without intermittency), scaling rates depend positively on the quantile $q$, increasing by more than 2% between the 95th quantile and the 99.5th quantile at the hourly time scale. By contrast, the model accounting for intermittency exhibits a much smaller sensitivity at the hourly scale. It consistently predicts a scaling rate of about 8% per degree Celsius, independently of the chosen quantile $q$. At the daily time scale, both models exhibit

similar sensitivities to the choice of the quantile. However, the model without intermittency fails to capture the temperature dependence due to the strong influence of intermittency at this scale (see previous Section). The model with intermittency predicts a more reasonable scaling rate of 5.7% to 6.9% per degree Celsius, slowly converging towards Clausius-Clapeyron scaling for larger values of $q$. The large spread for $q > 0.995$ can be interpreted as a sign that the selected time series are not long enough to accurately estimate rainfall quantiles above the $99.5^{\text{th}}$ quantile.

Perhaps one of the most striking features in Figure 11 is how sensitive the uncorrected scaling rates at the hourly time scale appear to be with respect to the choice of $q$. The shape of the trend could be used to support the idea that the largest rainfall extremes at small scales respond much faster to changes in temperature than expected from the Clausius-Clapeyron relationship. However, because the corrected scaling rates do not exhibit this trend, this increase with $q$ is likely to be a statistical artifact caused by intermittency.

To better understand this phenomena, it is important to look at how quickly intermittency levels change when going from one temperature class to another and how this rate varies with $q$. Intuitively, the average intermittency of rainfall extremes tends to decrease with $q$, resulting in stronger overall sensitivity to temperature. However, the decrease in intermittency from one quantile to another may not necessarily be uniform across all temperature bins. Typically, extremes at higher temperatures, which are more intermittent, will see their intermittency decrease at a faster rate than extremes at lower temperatures. These

non-uniform changes in intermittency between low and high temperatures can result in an apparent amplification of the scaling rate with temperature as we move towards larger $q$.

Figure 12 illustrates this scenario by showing the differences in observed rainfall amounts and intermittency for $q = 0.95$ and $q = 0.997$ versus temperature for the station in Fairhope, Alabama. The average intermittency decreases from 0.262 for $q = 0.95$ to 0.066 for $q = 0.997$. However, this difference is not uniformly spread across all temperature bins. As shown in

the bottom panel of Figure 12, intermittency at higher temperatures tends to decrease faster than at lower temperatures. This results in an amplification of extremes at higher temperatures with $q$ beyond that expected by Clausius-Clapeyron. One might argue that, in the end, it does not really matter whether the increase in total rainfall amount is caused by larger peak intensity or decreasing intermittency, or a combination of both, as long as the net rate of change is known. However, looking beyond rainfall totals, one also needs to take into account the fact that hydrological response is a combination of rainfall amount and

dynamics. Thus the interplay between peak intensity, duration and total rainfall amounts with temperature across scales is a



crucial factor to consider for flood risk analyses. In the end, storm water infrastructures need to be capable to deal with rainfall extremes across all relevant spatial and temporal scales. This requires more in depth knowledge of intermittency and rainfall dynamics as a function of atmospheric variables than is currently available.

## 5 Conclusions

Intermittency is a key feature controlling the variability of precipitation. Yet its effect is often poorly taken into account. The first main result of this study is that most rainfall extremes above hourly scales are intermittent in nature. For example, it is common for rainfall extremes at daily time scales to exhibit upward of 80% internal intermittency. For these reasons, peak intensity often turns out to be a rather weak predictor of total amounts compared with storm duration and dynamics.

The second important finding is that the current conceptual framework for studying the relationship between rainfall ex-
tremes and temperature based on Clausius-Clapeyron alone is too simplistic. Changes in extreme precipitation with temperature can not be reduced to a single number. Instead, there appears to be a seamless progression of changes, starting at the sub-hourly scales where rainfall extremes are predominantly controlled by variations in temperature up towards hourly, daily and weekly extremes which are increasingly dominated by intermittency. Temperature remains a crucial factor across all scales by controlling evaporation rates and the maximum moisture holding capacity of the air. But because intense storms with high
precipitation rates tend to run out of moisture more quickly, net effective changes in rainfall totals across scales remain hard to predict.

Despite decades of development, current numerical weather prediction models and climate simulations still lack the ability to reproduce realistic intermittent rainfall patterns, especially at sub-daily timescales where convective processes are the most important contributors to extremes. As a result, projections about the future of rainfall extremes are still very uncertain. Perhaps,
future developments might profit from the new scaling model proposed in this paper, allowing them to make a more in-depth analysis of how extremes react to changes in temperature across scales. New statistical metrics and diagnostic tools specifically designed to assess the realism of simulated intermittency patterns independently of total amounts might also prove useful (Schleiss and Smith, 2016). At the same time, more research is needed into the type of meteorological conditions capable of sustaining heavy rainfall intensities over a long period of time, including positive feedback mechanisms in mature storms,
modifications of large-scale moisture transport and spatio-temporal organization of storms across scales, none of which are fully understood yet.

Finally, note that while the present work only focused on temporal intermittency, the same approach could be used to study the internal intermittency of extremes aggregated over different spatial scales, for example by looking at how the fraction of dry pixels within a fixed area responds to changes in temperature. Also, since intermittency and temperature are not sufficient
to fully predict the response of heavy rainfall accumulations across scales, additional covariates like wind speed, dew point, pressure and vertical motion could be used in the analyses to further refine the models. Similarly, it might be worth to look at alternative intermittency metrics, like the fraction of the time the rainfall intensity exceeds a certain threshold or the temporal



variability of the rainfall rate within an extreme. The latter might offer a more detailed picture of internal storm variability than the simple binary rain/no-rain approach used in this paper.

*Data availability.* All data are freely accessible via anonymous ftp at: ftp://ftp.ncdc.noaa.gov/pub/data/uscrn/products/subhourly01.

*Competing interests.* The author has no competing interests.





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



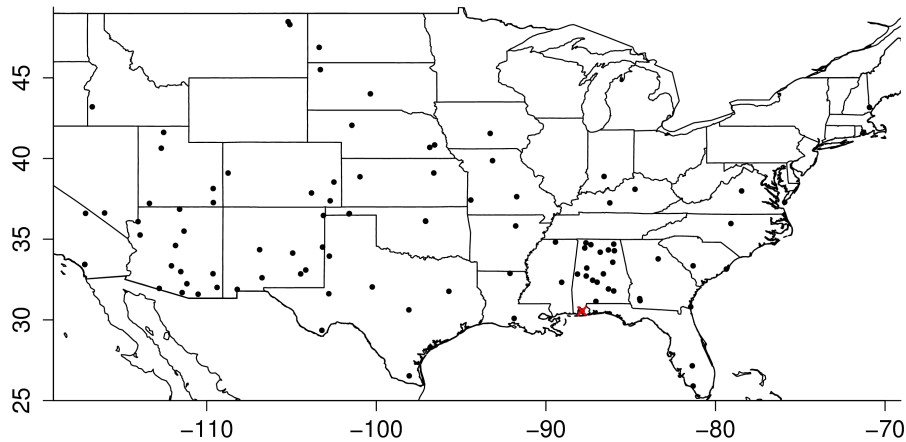

**Figure 1.** Map showing the location of the 99 selected USCRN stations. The red cross denotes station AL-Fairhope-3-NE which was chosen for illustration purposes.

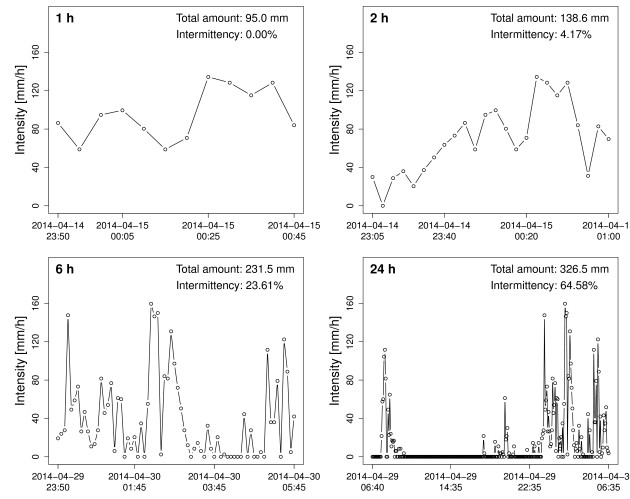

**Figure 2.** Highest rainfall accumulations recorded at AL-Fairhope-3-NE for 1 h, 2 h, 6 h and 24 h aggregation time scales.





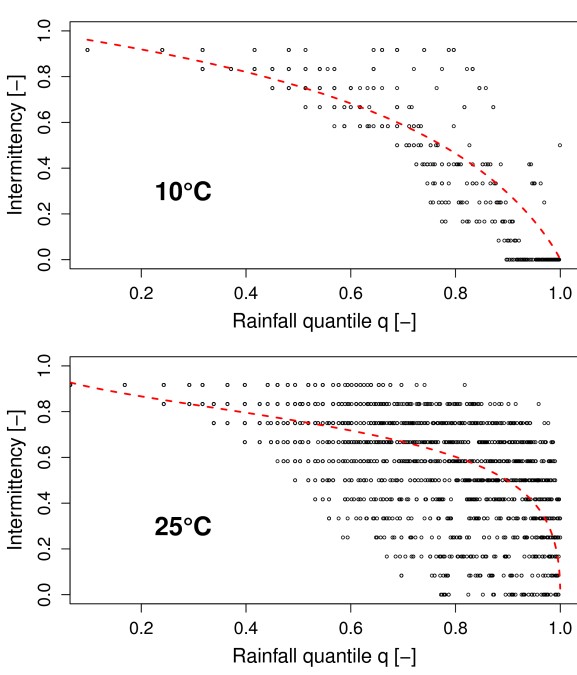

**Figure 3.** Logistic regression between intermittency and rainfall quantile at AL-Fairhope-3-NE for the 1 h time scale and two different temperatures (10°C and 25°C).





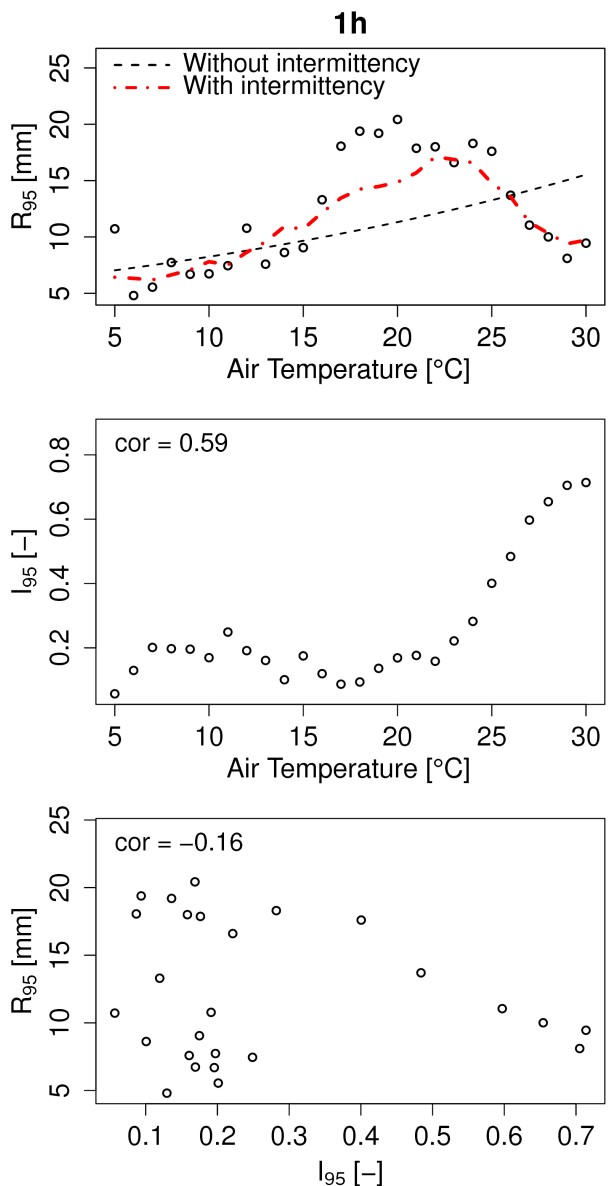

**Figure 4.** 95th quantile of rainfall amounts at the hourly time scale as a function of temperature and intermittency at AL-Fairhope-3-NE. Black dots denote sample estimates. The black and red lines represent the fitted scaling models given in Equations (9) and (11).





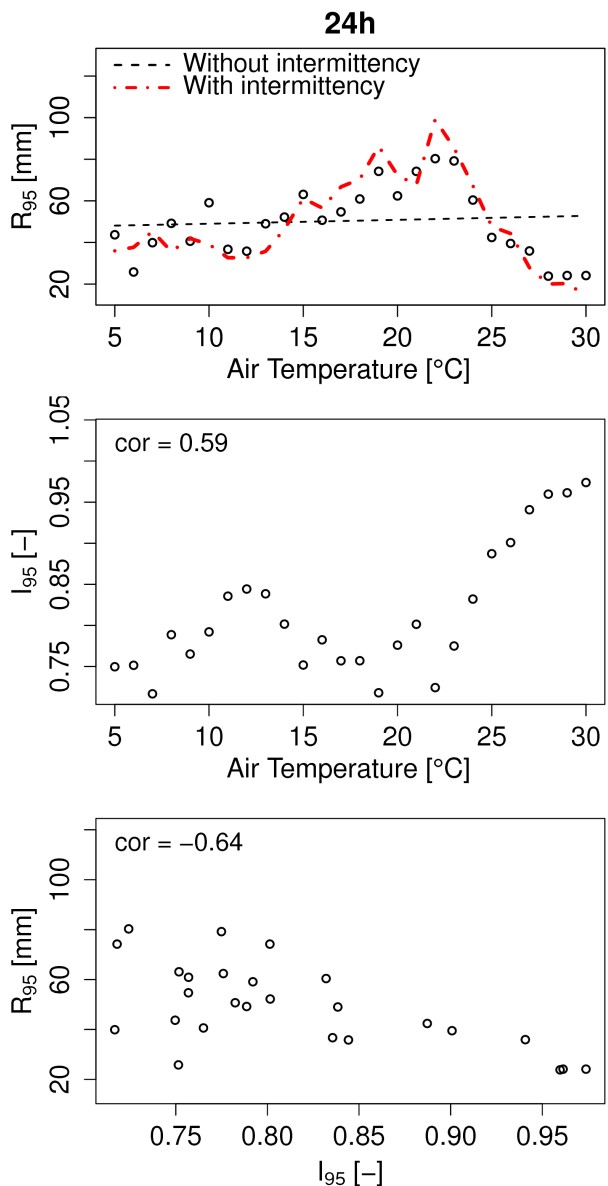

**Figure 5.** 95th quantile of rainfall amounts at the daily time scale as a function of temperature and intermittency at AL-Fairhope-3-NE. Black dots denote sample estimates. The black and red lines represent the fitted scaling models given in Equations (9) and (11).





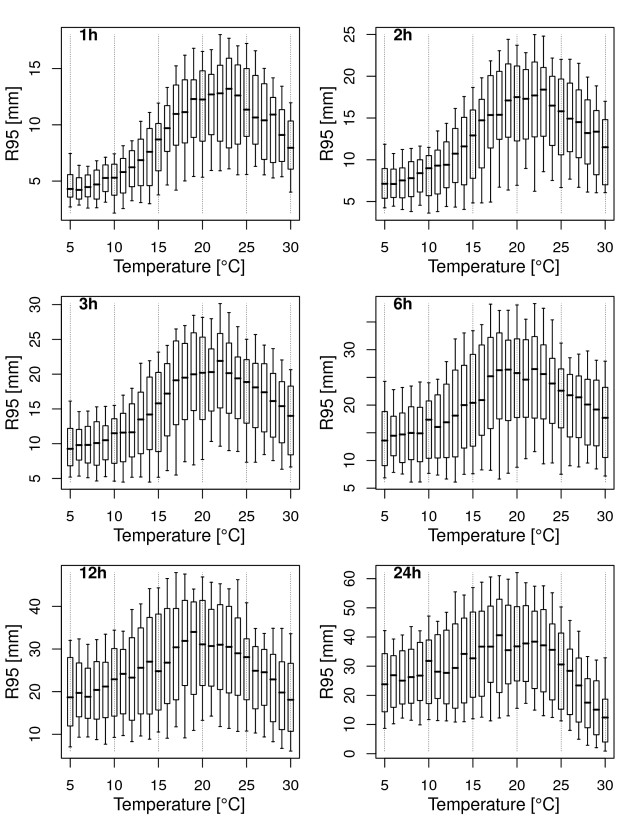

**Figure 6.** Box plots of 95th rainfall quantile vs temperature at 1 h, 2 h, 3 h, 6 h, 12 h and 24 h aggregation time scales.


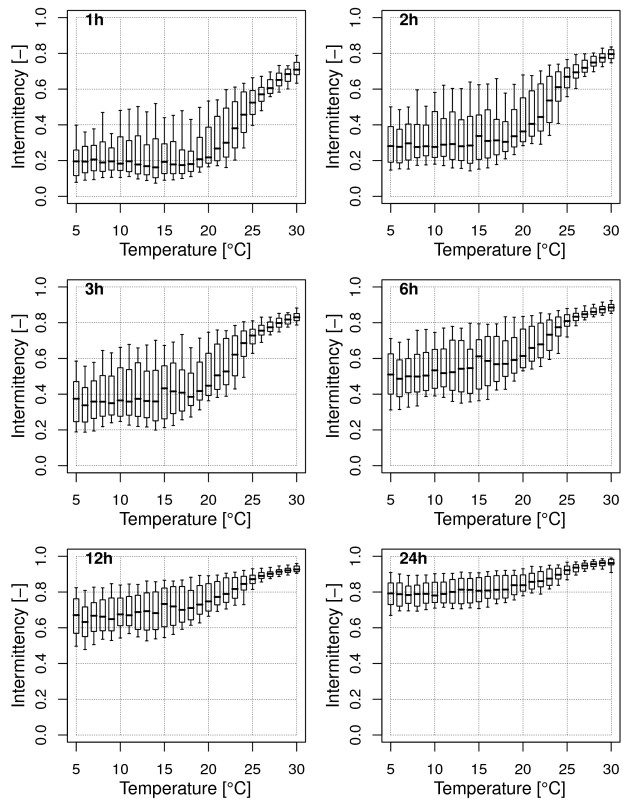

**Figure 7.** Box plots of internal intermittency vs temperature for 1 h, 2 h, 3 h, 6 h, 12 h and 24 h aggregation time scales.

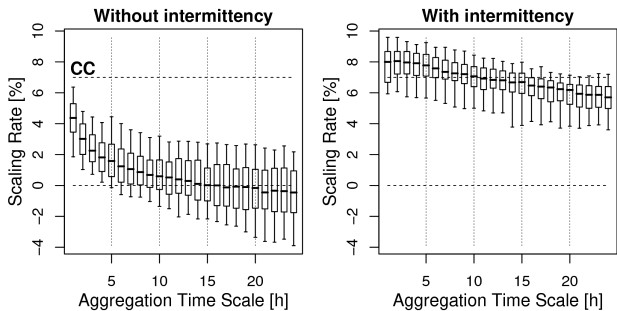

**Figure 8.** Box plots of estimated scaling rates of 95th rainfall quantile with temperature as a function of time scale. Each box plot shows the 10%, 25%, 50%, 75% and 90% quantiles of all 99 stations in the dataset.





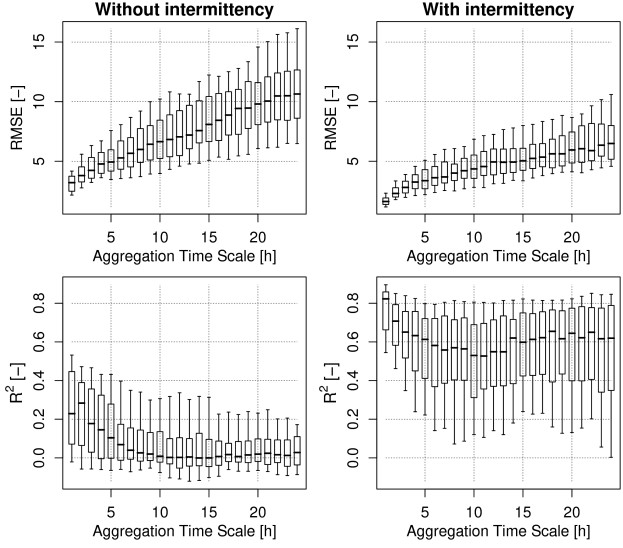

**Figure 9.** Box plots of root mean square error (RMSE) and coefficient of determination ($R^2$) of estimated 95th rainfall quantile as a function of scale. The model without intermittency is shown on the left and the model with intermittency on the right. Each box plot shows the 10%, 25%, 50%, 75% and 90% quantiles of all 99 stations in the dataset.

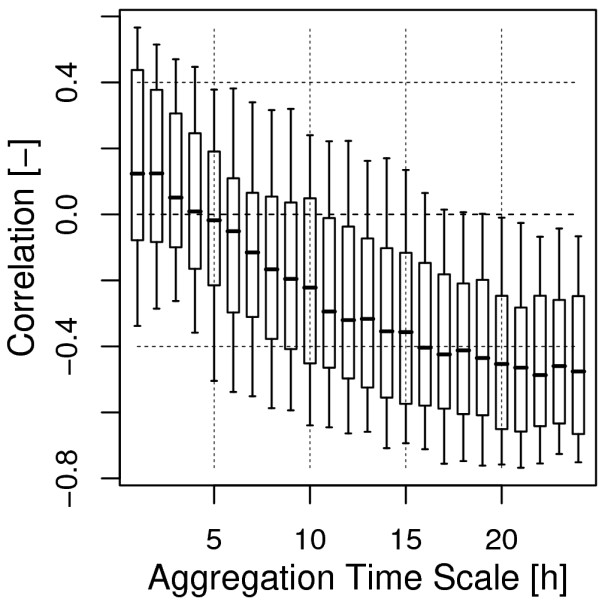

**Figure 10.** Box plots of Spearman rank correlation between 95th rainfall quantile and intermittency (over temperature) as a function of time scale. Each box plot shows the 10%, 25%, 50%, 75% and 90% quantiles of all 99 stations in the dataset.



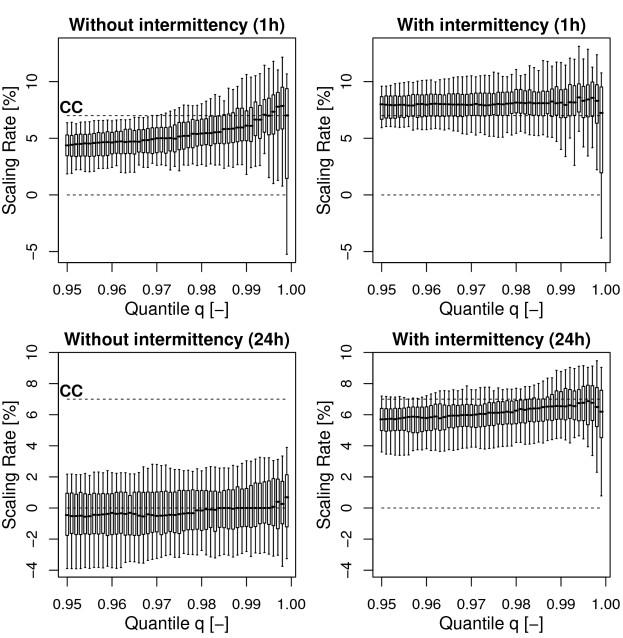

**Figure 11.** Box plots of rainfall scaling rates with temperature (5-30°C) as a function of the quantile $q$ for the 1 h and 24 h time scales. The model without correction for intermittency is shown on the left. The model with correction for intermittency on the right. Each box plot shows the 10%, 25%, 50%, 75% and 90% quantiles of all 99 stations in the dataset.





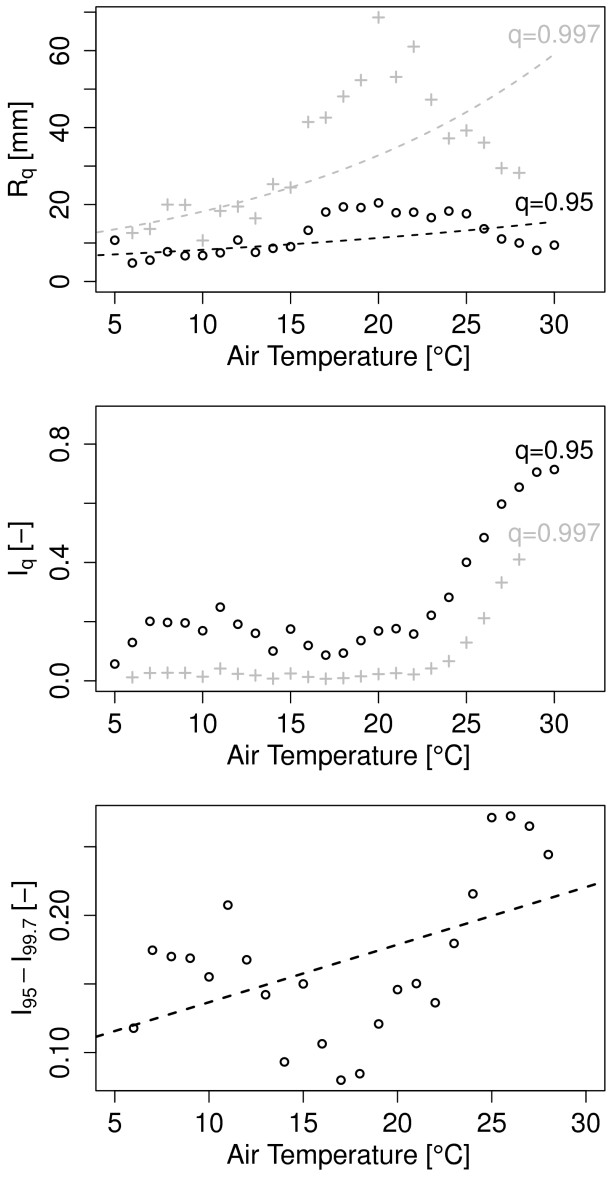

**Figure 12.** From top to bottom: rainfall quantiles, internal intermittency and changes in intermittency at the hourly time scale as a function of temperature and two different quantiles $q = 0.95$ and $q = 0.997$ for the station AL-Fairhope-3-NE.