# Peer review of "How intermittency affects the rate at which rainfall extremes respond to changes in temperature"

_Earth System Dynamics, 2018_

## Referee Comment (RC1) · Anonymous Referee #1 · 21 Feb 2018

This manuscript revisits the question of how rainfall intensity changes with temperature, considering this across different time scales, quantiles, and climate regions. The manuscript introduces the idea of quantifying intermittency of rain during a given time interval, and then applies a model that separates intermittency from thermodynamic controls on rainfall rates. This consideration of intermittency offers a novel way to interpret these intensity vs. temperature relationships and helps show that once intermittency is accounted for Clausius-Clapeyron scaling is preserved in many cases. This manuscript is a useful addition to the ongoing investigation of simple temperature controls on rainfall intensity.

My one major comment relates to the explanatory power of quantifying intermittency. Page 8 Line 15 suggests that intermittency is a "much simpler" explanation of changes

in scaling versus trying to explain shifts in dominant rainfall type. However, intermittency is not in itself a fundamental explanation. Temperature scaling of intensity has a basic physical principle at its root (the Clausius-Claperyon relationship). Measuring intermittency is a convenient way to quantify shifts in some underlying relationship, but the relationship between temperature and intermittency is largely empirical and not something known a prior from basic principles. There is no fundamental physical principle that describes intermittency.

I'm not questioning the value in quantifying intermittency and using it to isolate the thermodynamic scaling. But, I think the language in the manuscript should possibly be changed in places to better acknowledge that intermittency is not really the fundamental explanation, but just a convenient way to generally quantify the different dynamic controls limit atmospheric moisture availability. For instance, Page 12 Line 5 says "Intermittency is a key feature controlling the variability of precipitation". I would suggest it might be more appropriate to say something "Quantifying intermittency aids in identifying controls on variability of precipitation". There are several other places in the manuscript were this slight shift in presentation of intermittency might be warranted.

Minor Comments 1. In the abstract, the term intermittency should be defined more explicitly. At least for readers with less of a meteorology background, intermittency may be more familiar at longer time scales (dry periods and wet periods over weeks or months). So, the idea of considering intermittency over shorter time scales (hourly to daily) should be made clear at the beginning.

2. Page 3 Line 17 & Line 22; Line 17 indicates data is from the US CRN but line 22 indicates stations include Canada and Siberia. There is an inconsistency here given that that the US CRN only includes stations in the US and Figure 1 only shows U.S. sites.

3. Page 3 Line 29 states that aggregation was performed with overlapping time windows (shifted by 5 minutes). If overlapping windows were used, this would suggest

that each aggregate data point was not actually statistically independent , especially at longer time scales. For instance, if one had a 24 hour time window shifted by only 5 minutes, over 99% of the data from one window to the next would be the same. If this is what was actually done, there probably needs to be some accounting for the lack of independence of each data point. If this isn't what was done, then the text should be clarified.

4. Page 3 Line 32: "one or several missing values"? Shouldn't this just say "one or more missing values"?

5. Page 5 Line 3: maybe define lacunarity as used here?

6. Figure 12 What is dashed line in bottom panel?

7. Page 8 Lines 21 and 29 refer to changes in rainfall intensity per degree temperature while discussing Figures 4 and 5. If this can't be taken from the figure (I don't believe it can be), it could be helpful to be more clear that it comes directly from Eqn. 14 (especially at Line 29).

8. Page 8 Lines 9 to 14 This section mentions some different possible explanations of changes in scaling with temperature. As another explanation to possibly offer, I believe in some cases at high temperatures, there is moisture limitation at the land surface. The atmosphere might have capacity to hold moisture, but the land surface in the region has no moisture to give. Temperature merely becomes a proxy for the dryness of a given season.

---

## Author Comment (AC1) · 23 May 2018

**Response to Reviewer 1**

**Reviewer 1:** This manuscript revisits the question of how rainfall intensity changes with temperature, considering this across different time scales, quantiles, and climate regions. The manuscript introduces the idea of quantifying intermittency of rain during a given time interval, and then applies a model that separates intermittency from thermodynamic controls on rainfall rates. This consideration of intermittency offers a novel way to interpret these intensity vs. temperature relationships and helps show that once intermittency is accounted for Clausius-Clapeyron scaling is preserved in many cases. This manuscript is a useful addition to the ongoing investigation of simple temperature controls on rainfall intensity.

**Major comment:** My one major comment relates to the explanatory power of quantifying intermittency. Page 8 Line 15 suggests that intermittency is a "much simpler" explanation of changes in scaling versus trying to explain shifts in dominant rainfall type. However, intermittency is not in itself a fundamental explanation. Temperature scaling of intensity has a basic physical principle at its root (the Clausius Claperyon relationship). Measuring intermittency is a convenient way to quantify shifts in some underlying relationship, but the relationship between temperature and intermittency is largely empirical and not something known a prior from basic principles. There is no fundamental physical principle that describes intermittency. I'm not questioning the value in quantifying intermittency and using it to isolate the thermodynamic scaling. But, I think the language in the manuscript should possibly be changed in places to better acknowledge that intermittency is not really the fundamental explanation, but just a convenient way to generally quantify the different dynamic controls limit atmospheric moisture availability. For instance, Page 12 Line 5 says "Intermittency is a key feature controlling the variability of precipitation". I would suggest it might be more appropriate to say something "Quantifying intermittency aids in identifying controls on variability of precipitation". There are several other places in the manuscript were this slight shift in presentation of intermittency might be warranted.

**Response:** Thank you for the suggestion. I agree, these are two very different types of relationships and it's important to be very clear about that. Temperature and maximum rainfall intensity are linked by a direct physical relationship (described by Clausius-Clapeyron). The link between temperature and intermittency is less tangible. It only has an indirect physical interpretation in terms of the maximum rate at which precipitable water can be replenished through evaporation and advection from surrounding regions (which depends on actual physical factors like temperature, radiation and wind speed). I will go over the paper again to make sure this is clearly stated and change the language wherever necessary.

**Minor Comment 1:**
In the abstract, the term intermittency should be defined more explicitly. At least for readers with less of a meteorology background, intermittency may be more familiar at longer time scales (dry periods and wet periods over weeks or months). So, the idea of considering intermittency over shorter time scales (hourly to daily) should be made clear at the beginning.

**Response:** Yes, that would be a valuable addition. I will add a paragraph about small-scale intermittency and some references to relevant publications in the text. Possible references (with short summary or discussion) include:

- "Characterizing Multiscale Variability of Zero Intermittency in Spatial Rainfall", by Kumar, P. and Foufoula-Georgiou, E. (1994)

- "The Droplike Nature of Rain and Its Invariant Statistical Properties", by Ignaccolo, M. and De Michele, C. and Bianco, S. (2009).
- "New perspectives on rainfall from a discrete view" by De Michele, C. and Ignaccolo, M. (2013)
- "On the nature of rainfall intermittency as revealed  by different metrics and sampling approaches", by Mascaro, G. and Deidda, R. and Hellies, M. (2013).
- "Intra-event intermittency of rainfall: an analysis of the metrics of rain and no-rain periods" by Dunkerley, D. (2015)

**Minor comment 2:** Page 3 Line 17 & Line 22; Line 17 indicates data is from the USCRN but line 22 indicates stations include Canada and Siberia. There is an inconsistency here given that that the USCRN only includes stations in the US and Figure 1 only shows U.S. sites.

**Response:**
Yes, I see why this might be confusing. The USCRN network actually contains a few stations outside the U.S. But these did not satisfy the data requirements and were not considered in this analysis. I will revise the text accordingly to avoid any possible misunderstanding.

**Minor comment 3:** Page 3 Line 29 states that aggregation was performed with overlapping time windows (shifted by 5 minutes). If overlapping windows were used, this would suggest  that each aggregate data point was not actually statistically independent , especially at longer time scales. For instance, if one had a 24 hour time window shifted by only 5 minutes, over 99% of the data from one window to the next would be the same. If this is what was actually done, there probably needs to be some accounting for the lack of independence of each data point. If this isn't what was done, then the text should be clarified.

**Response:**
Yes, this is correct. The different values are not independent and there is no correction for this in the analyses. However, I don't think this is a major problem considering the lengths of the time series and the type of analyses I perform (please correct me if I'm wrong). The main reason I consider overlapping windows is to better account for the fact that the starting time of the measurement periods is arbitrary. Otherwise, results would depend on the choice of the starting time. In addition, the same independence problem arises when samples are taken over non-overlapping time windows, although in this case it mostly affects the smaller scales instead of the large ones. To avoid any confusion, I will add some details in the methods section to better explain the motivation behind this choice of overlapping windows.

**Minor comment 4:** Page 3 Line 32: "one or several missing values"? Shouldn't this just say "one or more missing values"?

**Response:**
Yes, the necessary correction will be made.

**Minor comment 5:** Page 5 Line 3: maybe define lacunarity as used here?

**Response:**
"lacunarity" should be understood in the sense of "how much of the time period is void of rain". The larger the lacunarity, the more the rainfall is concentrated in time. This is similar to the notion of fractals which only "fill" a certain fraction of the space over which they are defined.

**Minor comment 6:** Figure 12 What is dashed line in bottom panel?

**Response:**
The dashed line represents the fitted linear regression (using least squares) of the change in I95-I99.7 as a function of air temperature. I forgot to mention this in the text and will add it during the revision.

**Minor comment 7:** Page 8 Lines 21 and 29 refer to changes in rainfall intensity per degree temperature while discussing Figures 4 and 5. If this can't be taken from the figure (I don't believe it can be), it could be helpful to be more clear that it comes directly from Eqn. 14 (especially at Line 29).

**Response:** Sure, no problem. I'll add a sentence in the text to mention this.

**Minor comment 8:** Page 8 Lines 9 to 14 This section mentions some different possible explanations of changes in scaling with temperature. As another explanation to possibly offer, I believe in some cases at high temperatures, there is moisture limitation at the land surface. The atmosphere might have capacity to hold moisture, but the land surface in the region has no moisture to give. Temperature merely becomes a proxy for the dryness of a given season.

**Response:** Yes, I think this is a valid point. Actually, there have been several other studies mentioning this effect before. I will add it to the text and point to the literature for more details.

---

## Referee Comment (RC2) · Anonymous Referee #2 · 7 Jun 2018

This manuscript is a well deployed, aptly presented contribution to Earth System Science, even if more statistically than dynamically oriented.

A rare feature these days, this manuscript was a breeze to study: very clear and effective, ensuring a smooth assessment.

Overall, I recommend publication of this manuscript subject to minor revisions (very minor indeed).

In addition to the comments from the other reviewer, with whom I agree, and to which the author already gave a satisfactory answer, I would merely raise minor and technical notes:

My main concern pertains the largely descriptive nature of the study, notwithstanding

the technical merits of the analysis and the relevance of the presented developments for modelling purposes.

Minor remark:

As the title itself indicates, this is a study of "how", not necessarily of "why". Therefore, I would not require the authors to delve into the fundamental mechanisms behind the features being identified and analysed as that would make for a different study. Even so, it would be a nice addition to the paper to complement what is currently a good discussion with further remarks of mechanistic nature (a couple of sentences should suffice). This way, the physically oriented readers among the ESD community would be even more appreciative.

Technical/notational remarks:

Equation 1: the opening and closing brackets in the Probability operator should be the same - either $[]$ or $()$.

Equations 2 and 3: 'ni' should actually read $n_i$ (subscript $i$ as it is an index), otherwise it will appear as if $n$ is multiplying by $i$.

Page 5, last line: The variable $q$ should come in math type (italicised) (Same remark on Page 6, line 14)

Equation 4: The Expectation operator should be identified as such in the text.

Equations 5, 7: The hierarchy of brackets would be recommended: $[()]$ rather than $(())$.

Thank you!

---

## Author Comment (AC2) · 7 Jun 2018

**Response to Reviewer 2**

**Reviewer 2:** This manuscript is a well deployed, aptly presented contribution to Earth System Science, even if more statistically than dynamically oriented. A rare feature these days, this manuscript was a breeze to study: very clear and effective, ensuring a smooth assessment. Overall, I recommend publication of this manuscript subject to minor revisions (very minor indeed). In addition to the comments from the other reviewer, with whom I agree, and to which the author already gave a satisfactory answer, I would merely raise minor and technical notes: My main concern pertains the largely descriptive nature of the study, notwithstanding the technical merits of the analysis and the relevance of the presented developments for modelling purposes. As the title itself indicates, this is a study of "how", not necessarily of "why". Therefore, I would not require the authors to delve into the fundamental mechanisms behind the features being identified and analysed as that would make for a different study. Even so, it would be a nice addition to the paper to complement what is currently a good discussion with further remarks of mechanistic nature (a couple of sentences should suffice). This way, the physically oriented readers among the ESD community would be even more appreciative.

**Response:** Thank you for the kind and encouraging words. Indeed, the paper could benefit from some additional discussion about the physical mechanisms responsible for producing intermittency in rainfall and how these relate to the statistical analyses presented in this paper. I will add a new paragraph about that in the discussion, together with some references to the literature. As a start, I could use the paper by Neelin et al. (2017) *"Global warming precipitation accumulation increases above the current-climate cutoff scale"*, especially the part describing the link between the local fluctuations in atmospheric moisture during a rain event and the dynamics of precipitation accumulations on the ground. I'll also add a few words about small-scale intermittency and its link to atmospheric turbulence.

**Technical/notational remarks:**
- Equation 1: the opening and closing brackets in the Probability operator should be the same - either [] or ().
- Equations 2 and 3: 'ni' should actually read n i (subscript i as it is an index), otherwise it will appear as if n is multiplying by i.
- Page 5, last line: The variable q should come in math type (italicised) (Same remark on Page 6, line 14)
- Equation 4: The Expectation operator should be identified as such in the text.
- Equations 5, 7: The hierarchy of brackets would be recommended: [()] rather than (()).

**Response:** Thanks! I'll correct these during the revision.

---

## Author Response (AR1)

**How intermittency affects the rate at which rainfall extremes respond to changes in temperature**

Final response to reviewers and editor
Marc Schleiss

I would like to thank the reviewers and the editor for their valuable comments and suggestions for improving the quality of this paper. I have taken each comment into consideration and performed all necessary changes (with the exception of a few minor technical issues explained below). In the following, please find a point-by-point description of the main changes made during revision together with references to the corresponding page numbers and sections.

**Main changes to the paper:**

- I went over the whole paper again to make sure the term "intermittency" was clearly defined and that the difference between intermittency (i.e., a statistical summary of the lacunarity of rain) and the actual physical processes responsible for it (e.g., evaporation, advection, convergence and formation of precipitation) was clear.

- In the introduction, I added a sentence about the difference between large-scale and small-scale intermittency with some new references to the literature:

*"Indeed, beyond a few hours of aggregation time scale, total rainfall amounts often turn out to be more correlated to storm duration and intermittency rather than peak rainfall intensity (e.g., Azad and Sorteberg, 2017; Lamjiri et al., 2017). And while the discrete, episodic nature of precipitation may be most apparent at larger scales (e.g., days, weeks or months), its effects can be observed down to the microscale (e.g., Kumar and Foufoula-Georgiou, 1994; Ignaccolo et al., 2009; De Michele and Ignaccolo, 2013; Mascaro et al., 2013)."*

- I added more details in Section 4.1 about the physical mechanisms responsible for producing intermittency in rain. In the same section, I also added the additional explanation suggested by Reviewer 1 for why the scaling of extremes might change above 22-23 degrees Celsius:

*"Another explanation could be the existence of strong moisture limitations in the regions surrounding the rainfall and upwind thereof. The atmosphere might have capacity to hold more water at higher temperatures, but the land surfaces have no additional moisture to give, causing the relationship between temperature and rainfall extremes to change. The key parameter in this case is the rate at which new precipitable water can be evaporated and brought in from surrounding regions, which increases with temperature but will be limited by advection velocities and moisture availability at nearby land surfaces. A simple calculation of daily mean evaporation rates with temperature using the approximation provided by Linacre (1977) confirms this hypothesis, showing that although mean evaporation rates increase steadily with temperature, the rate of increases slows down at higher temperatures. Even in cases of unlimited moisture supply, evaporation rates remain small compared with precipitation rates. Thus, once all the water in a column of air has been rained out, the dominant factors controlling precipitation totals at scales beyond one hour are likely to be dynamical in nature. Intermittency, although it is not a physical quantity, can be viewed as a summary statistic of the combined effect of all dynamical processes at work in rainfall. As such, it can help better understand*

*the response of rainfall extremes to changing temperatures beyond simple Clausius-Clapeyron scaling."*

- I changed the text in the data and methods section to clarify the difference between the full USCRN dataset (which contains stations outside of the U.S.) and the subset used for analysis (which only contains data in the U.S.).

*"The full weather station network consisted of 232 different stations spread across the United States, Canada and Siberia  However, only a small subset of these stations were kept for the analysis. Specifically, only the time series with at least 20 valid positive rainfall values in at least 20 different temperature classes between 5 and 30 degrees Celsius at the 24h aggregation time scale were kept. This drastically reduced the number of stations, from 232 to 99. A map with the 99 stations satisfying all these criteria is shown in Figure 1."*

- In the data and methods section, I added a sentence explaining why aggregation was performed over overlapping time windows.

*"The main reason for using overlapping time windows during aggregation was to better account for the fact that the starting time of an aggregation time period is arbitrary. By contrast, non-overlapping time windows would have resulted in many large precipitation accumulations to be missed."*

- In section 4.1 (pages 9-10) I added information about the uncertainty on the estimated scaling rates, as suggested by the editor. Average uncertainties on estimated scaling rates were 1.8 - 2.7% for the model without intermittency and 1.3 - 1.7% with intermittency. However, individual uncertainties on scaling rates for selected stations and time scales can be as high as 5%, as illustrated by the station *FL-Everglades-City-5-NE*. I did not perform any formal statistical testing as I felt this was not real needed here to get the message: there are obvious differences between the two methods at larger scales and not so much at the smaller scales.

*"In the model without intermittency, scaling rates rapidly decrease with Delta t from approximately 4.37% at the hourly time scale to -0.45% at the 24h scale. The average uncertainty affecting the estimated scaling rates at a given time scale (among all stations) is between 1.8% and 2.7% and increases with Delta t. […] After correcting for intermittency, the effect of temperature becomes visible again and results are much closer to what can be expected from the Clausius-Clapeyron relationship. Still, there appears to be a small decrease of the scaling rate with Delta t after correction for intermittency from 8.0% at the hourly scale to 5.70% at the 24h scale. The latter however, can be explained by the relatively small sample sizes and is well within the range of uncertainty (average uncertainty of 1.3 - 1.7% per time scale, increasing with Delta t)."*

*"The stations with the strongest scaling rates overall (both at the hourly and daily time scales) were FL-Sebring-23-SSE (12.96% +/- 3.4%  without intermittency and 14.70% +/- 1.74%  with intermittency) and FL-Everglades-City-5-NE (12.42% +/- 5.15% respectively 13.04% +/- 2.41%), both situated in a humid tropical climate famous for large and intense warm season thunderstorms"*

- Following up on the suggestion by the editor, I added an additional sentence at the end of the conclusion mentioning the necessity to perform more in-depth analysis of the joint and conditional distribution of the triplets of rainfall (R), temperature (T) and intermittency (I) for all rainfall quantiles, and not just the extremes:

*"A more detailed and systematic analysis of the joint probability distribution of (R,T,I) and pairwise conditional density functions for all values of rainfall accumulations (and not only for the upper quantiles) might also be beneficial to better understand how rainfall amounts, temperature and intermittency are linked across scales."*

**Technical/notational remarks:**

All suggested changes have been performed with the exception of the following:

- Equations 2 and 3: 'ni' should actually read n i (subscript i as it is an index), otherwise it will appear as if n is multiplying by i.

*Response: There is no need for a subscript here as this is indeed a multiplication.*

- Equations 5, 7: The hierarchy of brackets would be recommended: [()] rather than (()).

*Response: Square brackets are not recommended here. For functions, it's better to use ().*

- While the author presented the relations for all the 99 stations using boxplots, it would be better if they are presented in maps to understand the spatial variations better. Perhaps, as discussed in section 4, the author can divide the scaling exponents under different temperature gradients and then maps the exponents.

*Response: Maps containing the results of the 99 stations would have been possible. I tried this in some of the earlier versions of the paper (before initial submission). However, I felt it was hard to interpret these (due to strong variations in space, scales and temperatures) and to draw any strong conclusion from them. The boxplots on the other hand provide a much clearer summary of the data.*

- Editor comment: figures have to be improved to make them presentation quality.

*How exactly? Some suggestions for improvement would be welcome.*